# Muscle-Specific Endurance of the Lower Back Erectors Using Electrical Twitch Mechanomyography

**DOI:** 10.3390/jfmk4010012

**Published:** 2019-01-27

**Authors:** Kevin K. McCully, Caio Moraes, Sahil V. Patel, Max Green, T. Bradley Willingham

**Affiliations:** 1Department of Kinesiology, University of Georgia, Athens, GA 30602, USA; 2AU-UGA Medical Partnership, Athens, GA 30606, USA

**Keywords:** muscle fatigue, fatigability, neuromuscular electrical stimulation, humans, near infrared spectroscopy, reproducibility

## Abstract

Lower back pain is a common symptom potentially associated with skeletal muscle dysfunction. The purpose of this study was to evaluate endurance in the lower back muscles of healthy participants using accelerometer-based mechanomyography. Methods: Young healthy subjects (N = 7) were tested. Surface electrodes and a tri-axial accelerometer were placed over the erector spinae muscle along the T11–L1 Vertebrae. Stimulation was for 3 min each at 2, 4, and 6 Hz, and changes in acceleration were used to calculate an endurance index (EI). Reproducibility of the endurance index measurements was tested on two separate days. Wrist flexor and vastus lateralis muscles were tested for comparison. Near Infrared Spectroscopy (NIRS) was used to measure muscle oxygen levels (O_2_Hb) (N = 5). EI was 70.3 + 13.4, 32.6 + 8.4, and 19.2 + 6.2% for 2, 4, 6 Hz, respectively. The coefficients of variation were 9.8, 13.9, and 20.3% for 2, 4, 6 Hz, respectively. EI values were lower in the erector spinae muscles compared to the arm and the leg (*p* < 0.05). O_2_Hb values were 86.4 + 10.9% at rest and were 77.2 + 15.5, 84.3 + 14.1, and 84.1 + 18.9% for 2, 4, 6 Hz, respectively (*p* > 0.05, all comparisons). An endurance index can be obtained from the lower back erectors muscles that is reproducible and not influenced by voluntary effort or muscle oxygen levels.

## 1. Introduction

Chronic lower back pain is a significant medical problem which affects 100 million adults [1]. One of the limitations to treating lower back pain is the need for better assessment tools, including assessments of muscle fatigue [1,2]. While fatigue of lower back muscles has been implicated or associated with lower back pain, the fatigability of these muscles is difficult to assess. Fatigue of the lower back muscles has been evaluated using performance tests [3], changes in muscle electromyography signals [4,5], and by other innovative approaches [6,7]. However, all of these previous approaches are considered to have methodological limitations [8]. These limitations include requiring maximal muscle activation in an expensive ergometer and reliance on changes in EMG signals to reflect muscle fatigue.

An endurance protocol has been recently developed that may have utility for studying fatigue in lower back muscles. A triaxial accelerometer is used to measure the acceleration (aMMG) of skeletal muscles produced by low frequency surface neuromuscular electrical stimulation. As the muscle fatigues, acceleration declines. The remaining acceleration at the end of the stimulation is used to calculate an endurance index (EI). Previous studies have shown that aMMG changes in muscle acceleration correlate well with changes in muscle force production [9,10,11,12]. 

This study evaluated the reproducibility of the endurance index measured by aMMG as a method of assessing muscle fatigue in the erector spinae muscles of the lower back. Endurance indexes were assessed on two occasions, and the coefficient of variation and interclass correlation coefficients were determined. In addition, endurance indexes were assessed for comparison on the forearm and vastus lateralis muscles. Finally, muscle oxygen saturation was measured using near infrared spectroscopy (NIRS) during electrical stimulation to determine if contraction-induced ischemia occurred during the determination of the endurance indexes. 

## 2. Materials and Methods

### 2.1. Participants

Eight healthy participants (5 males and 3 females) participated in this study. Participants were 22.9 + 2 years old, 1.8 + 0.1 m tall, and 69.8 + 14.9 kg of body weight. Participants had no prior history of lower back pain. The study (Protocol ID#study0000001, 17 May 2013) was approved by the Institutional Review Board at the University of Georgia (Athens, Ga). All the subjects gave informed, written consent before testing.

### 2.2. Experimental Design

Each participant was tested on three separate days. On one day, the endurance index was determined for the erector spinae and vastus lateralis muscles. On another day, the endurance index was determined on the erector spinae and the forearm/wrist flexor muscles. On another day, oxygen saturation was measured in the erector spinae muscles. The order of testing of the three days was varied for different participants, and all experiments were completed within two weeks for each participant. 

### 2.3. Endurance Index Measurements

For the erector spinae muscle testing, the subjects were seated in a chair and leaned forward with their chest against an adjustable table and their head resting on a pillow placed on top of their folded arms (Figure 1). The position was standardized at a 60-degree torso angle. The upper body was also strapped to a padded board to minimize torso movement during the endurance measurements. Two electrodes (5 × 5cm) were placed with a 4 cm gap on the exposed lower back lateral to the T11 and L1 vertebrae on the belly of the left erector spinae (Figure 1). The muscles were stimulated using current levels that ranged from 25 to 35 mA with pulse durations/intervals of 200/50 us. A wireless tri-axial accelerometer (WAX3, Axivity, Newcastle upon Tyne, UK) was used to measure the acceleration of the muscle twitch contractions. 

The endurance protocol consisted of electrical stimulation at 2 Hz, 4 Hz, and 6 Hz (3 min each) with 5 s of no stimulation between each stage [13]. The accelerometer measured the surface oscillations resulting from the muscle twitches via wireless Bluetooth transmission. A resultant acceleration (Ar) was calculated from all three axes. Peak to peak (p-p) analysis was used to determine the magnitude of acceleration measured during each contraction. The endurance index was calculated as the percentage of acceleration at the end of the fatigue test compared to the initial peak value. The data analysis was performed using custom written routines in MATLAB R2014b (MathWorks Inc., Natick, MA, USA) [13].

For the forearm and vastus lateralis muscles, the accelerometer was placed in the center of the belly of the muscles and the stimulation electrodes were placed proximally and distally on the same muscle. The current used and stimulation protocol were the same as for the erector spinae muscles. 

### 2.4. Oxygen Saturation During the Endurance Index Test

For the erector spinae muscles, the participant was strapped into a lower lumbar extension machine (Cybex Eagle, Rosemont, IL, USA). A near infrared spectroscopy (NIRS) device (Portamon, Artinis Medical Systems, Einsteinweg, The Netherlands) was secured to the subject using an elastic band. The location was along the T11–L1 vertebrae in approximately the same location as the accelerometer placement. Blood oxygenation (O_2_Hb), deoxygenation (HHb), and blood volume were collected at a sampling rate of 10 Hz with separation distances of 3 and 4 cm. The same 2, 4, and 6 Hz protocols were performed, followed by a maximal voluntary isometric back extension contraction held for 90 s. Data collection continued for 90 s after the back extension to allow for baseline values to reestablish. Oxygen saturation (difference signal of O_2_Hb and HHb) was recorded at rest during the last 30 s of each stimulation level, at the lowest point of the isometric contraction, and at the highest point after the isometric contraction. The values of oxygen saturation for rest and the stimulation levels were calculated as a percentage of the range between the low values during the isometric contraction (assumed to be 0% oxygen saturation) and the highest values during reactive hyperemia (assumed to be 100% oxygen saturation) [14]. 

### 2.5. Analysis

Measures of the endurance index between tests were compared using one-way analysis of variance (ANOVA). Post-Hoc Bonferroni corrections for multiple comparisons were used to identify differences in the endurance index values at each frequency of stimulation. Interclass correlation coefficients (ICC) were also calculated. Oxygen saturations during different levels of stimulation were compared using repeated measures ANOVA. Data are presented as means with standard deviations (SD). Statistical significance was accepted with a *p* value < 0.05. 

## 3. Results

Positioning of the stimulation electrodes and a representative response of acceleration for the erector spinae muscles is shown in Figure 1. 

The endurance index values for the vastus lateralis, wrist flexor, and erector spinae muscles are show in Figure 2A. There was a main effect of the muscle on the EI values (F = 6.64, *p* < 0.001). Multiple comparisons between muscles were all significant, with the erector spinae muscle having *p* values < 0.001 for both limb muscles, and the comparison between the Vastus lateralis and Wrist flexors having a *p* value = 0.031. Comparisons in the erector spinae muscles between two tests on separate days are shown in Figure 2B. There was no effect of the day on the EI values (F = 0.188, *p* = 0.831). The coefficient of variation values for the erector spinae muscles were 9.8, 13.9, and 20.3% for 2, 4, and 6 Hz stimulation, respectively. The interclass correlation coefficients were 0.80, 0.86, and 0.59 for 2, 4, and 6 Hz stimulation, respectively. 

Oxygen saturation in the erector spinae muscle dropped significantly with isometric back contractions, as shown in the representative trace in Figure 3A. However, there were no declines in oxygen levels during electrical stimulation at any frequency compared to the resting values as shown in Figure 3B. 

## 4. Discussion

This study used muscle acceleration and twitch electrical stimulation to evaluate muscle endurance in the lower back erector muscles. The primary findings in this study were that an endurance index can be determined for the erector spinae muscles at three different stimulation intensities, that EI values have reasonable reproducibility, and that the EI values for the erector spinae in asymptomatic participants are lower than the EI values from muscles in the leg and arm. In our study, the arm muscle had lower EI values compared to the leg muscles, which was consistent with previously published values for mitochondrial capacity [14,15]. There is currently no data on mitochondrial capacity for the erector spinae muscles; however, the lower endurance observed in the erector spinae muscles could be related to lower levels of activity and oxidative capacity compared to the arm and leg muscles. The average EI value at 6 Hz for the erector spinae is even lower than that reported for the trapezius muscle (42% for 6Hz) in a previous study [16]. 

In previous studies, back muscle fatigue or endurance has been measured using repeated back extensions and a back muscle ergometer (1S6216, 2). The most commonly used test is the Biering–Sorensen test [2]. The Biering–Sorensen test involves a continuously held horizontal back extension and has been considered the most reproducible of tests of back endurance [17]. The ICC values for this test ranged from 0.88 for people with current nonspecific back pain, 0.77 for people with a history of nonspecific back pain, and 0.83 for people without back pain [17]. These values are similar to the ICC values found in our study. The reproducibility of the EI values in this study was between 10–20%. This was similar to the reproducibly found for the trapezius muscles using a similar endurance protocol [16]. The EI values and the ICC values for 2 and 4 Hz were agreeable, and those tests could be considered to have acceptable reproducibility. The lower reproducibility found at 6 Hz may reflect the low EI values (17–19%) seen for this stimulation rate. We found the back position difficult to keep consistent, and this may have also contributed to the lower reproducibility seen at 6 Hz. The advantages of our endurance index are that it does not require a maximal voluntary effort, it does not involve the use of “raters” to evaluate when the test is finished, and it is not influenced by factors such as the height and weight of the participant. 

We chose to use accelerometer-based mechanomyography (aMMG) to evaluate muscle endurance [13]. This approach has been used successfully on limb muscles [18,19] and muscles in the upper back [16]. Studies of the lower back muscles have reported changes in surface muscle electromyography signals and force development as indices of fatigue/endurance [4,5]. EMG has been previously used to monitor fatigue in various muscles, but measures of muscle activation are an indirect measure of muscle performance and do not always correlate with force development or movement. Force development by muscles in the lower back has been evaluated using specialized ergometers [20,21]. However, these ergometers and other tests of lower back muscle function (like the Biering–Sorensen test) measure muscle fatigue using sustained or repeated isometric contractions. Voluntary, isometric contractions may be limited in their ability to specifically characterize muscle function in the lower back. By using aMMG to measure movement rather than force, we were able to measure the strength of muscle contractions without the use of ergometers. The twitch endurance protocol also avoided using high-force voluntary contractions. In addition, muscle twitches can be considered shortening contractions, which are known to have higher rates of cross-bridge turnover and ATP-use than isometric contractions. The increased metabolic cost of shortening contractions makes twitch muscle endurance tests more likely to reflect the metabolic capacity of the muscle being tested. 

One of the concerns with any fatigue test related to muscle metabolism is that the muscle contractions may limit oxygen delivery [22]. If blood flow is limited, measures of endurance will reflect the extent of the limitations in blood flow rather than the metabolic capacity of the muscle. For example, the Biering–Sorensen test and the use of specialized ergometers for the lower back have involved sustained maximal isometric contractions. In our study, we found a sustained isometric contraction lowered oxygen saturation levels. Because we have found that sustained isometric contractions lower oxygen levels to similar values as cuff-induced ischemia, we used a sustained contraction to calibrate our NIRS signals. Compared to a sustained isometric contraction, the electrical stimulation twitch protocols produced little or no decrease in oxygen levels compared to the resting values. Several previous studies have measured changes in oxygen levels to the Biering–Sorensen test [23,24,25]. These studies found sustained isometric exercise lowered oxygen levels, although the results seemed to vary between subjects. It is difficult to compare our oxygen levels to theirs because we used longer separation distances for our light source and detector (4.5 mm compared to either 2.0 or 3.5 mm), and wider separation distances result in greater light penetration depth to reach more muscle and less superficial tissue. In addition, we normalized our NIRS signal to the maximal value rather than reported the NIRS values in optical density units. Overall, the difference in oxygen levels during twitch stimulation and sustained isometric contractions found in this study suggests that impaired oxygen delivery due to decreased muscle blood flow does not contribute to the EI measured during twitch contractions. 

One of the limitations to our study was the use of healthy, younger subjects without a history of lower back pain or weakness. This was done as a first step to evaluate the use of the EI test on the back muscles. Future studies should evaluate the utility of the EI measurements in patient populations. Because involuntary and submaximal stimulations are used, the test has the potential to be applied to a wide variety of patient populations. 

## 5. Conclusions

We found that the erector spinae muscles have lower endurance index values compared to leg and arm muscles in young healthy participants without a recent history of back pain. The EI values were reasonably reproducible, especially for the 2 and 4 Hz stimulation rates. The stimulation protocol showed no evidence of limiting oxygen delivery, and thus, the test should reflect oxidative metabolism rather than contraction-induced limited blood flow. The twitch EI test has the potential to evaluate patients with lower back pain and other clinical populations. 

## Figures and Tables

**Figure 1 jfmk-04-00012-f001:**
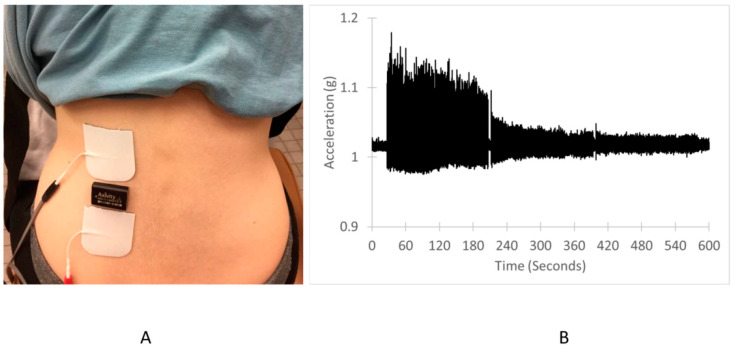
(**A**) Experimental set-up for the lower back fatigue test: The stimulation electrodes are white and the accelerometer is black, located between the electrodes, and (**B**) representative acceleration values (resultant vector) for a participant. This participant showed a large decline in acceleration with 4 and 6 Hz stimulation.

**Figure 2 jfmk-04-00012-f002:**
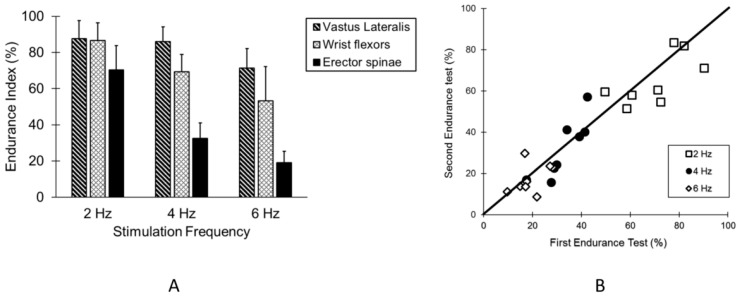
(**A**) The average EI values for the leg, arm and back muscles. The values are means and SD. Significant differences between back muscle EI and leg muscles are indicated by *, and significant differences between arm and back muscles are indicated by #; (**B**) the endurance index values for the erector spinae muscles for each subject on two separate days.

**Figure 3 jfmk-04-00012-f003:**
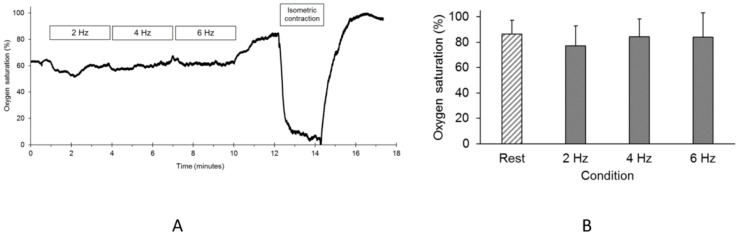
(**A**) A representative example of NIRS-measured oxygen levels in the erector spinae muscles. The periods of electrical stimulation and the time the subject performed a maximal isometric back contraction are indicated; (**B**) the average oxygen saturation values (O_2_Hb) for the back muscles at rest and during electrical stimulation. The units are as a percentage of the signal change during a sustained isometric contraction. The values are means and SD.

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
