# Peer review of "Muscle-Specific Endurance of the Lower Back Erectors Using Electrical Twitch Mechanomyography"

_jfmk, 2019, doi:10.3390/jfmk4010012_

Reviewer 1 Report

This study extends our understanding that aMMG and EI% can assess skeletal muscle specific endurance.  Furthermore, the study provides evidence that a simple test to assess skeletal muscle endurance may be useful technique to assess the contribution of skeletal muscle to a medical condition that impacts most adults.

Minor Issues:

Several miss-spelled words

Line 13: "health" should be "healthy"

Line 22: Add "muscles to erector spinae"muscles" compared..

Line 25: "of" should "or"

Line 42: Add "skeletal" to acceleration of "skeletal" muscle movement

Line 48: Change "endurance indexes were performed" to "endurance indexes was assessed" contractions were performed, EI% assesses characteristics about those contractions.

Line 50: Change "were determined" to "was assessed" 

Line 59: Remove "were" from "were gave"

Line 60: Add "." to end of sentence

Lines 64 and 65: please indicate that EI% aMMG was performed to assess changes in oxygen saturation.

Line 85: should "movement" be replaced by "acceleration" since it is this characteristic that was assessed by the accelerometer?

Line 85: Please add a reference for calculation methods.

Line 100: Replace "simulation" with "stimulation".

Line 102: reference ?

Line 127: should electrical stimulation be replaced by aMMG?

Line 139: replace "that" with "than"

Line 147: Correct reference "1S6216"

Line 168: Replace "test" with "tests"

Line 170: remove "and" from "contractions and may be"

Line 179: Reference?

Line 184: change "low" to "lower"

Line 205: change "reasonable" to reasonably"

Larger issue:

Use of a One-way ANOVA to assess EI% but repeated measures ANOVA to assess Oxygen Saturation.  It is not clear why a repeated measures ANOVA would be used in one instance with the same design but not in another.  Furthermore, I would recommend a two-way repeated measures ANOVA for the aMMG analysis since participants participated on 2 separate occasions (day) and each test consisted of 3 periods of stimulation (time).  Use contrasts with Bonferonni Correction to test for specific differences.

A two-way ANOVA with 1 repeated factor (time) should be used to compare the three different muscle groups for figure 2a.  Again use contrasts to assess specific differences.

Author Response

We appreciate the reviewers careful reading of our manuscript.  The suggested changes have been made unless otherwise indicated.

Line 64-65.  We are not sure about this comment.  Endurance is measured with acceleration and oxygen levels by NIRS.  This is described in the later two paragraphs. 

Line 127.  We feel this sentence should remain unchanged.  It is the muscle activation caused by the stimulation that might change oxygen levels. 

We have revised our statistical analysis, hopefully to address the issues presented by the reviewer.  the new analysis does not change the interpretation of the results, but we feel does improve the study. 

Reviewer 2 Report

Well written manuscript: concise but with all necessary details covered; cleanly organized; justification, analysis and discussion answers all questions reader may have (about why authors chose aMMG over traditional EMG or Biering-Sorensen test).

I only spotted a few typo/misspellings: page 5, line 184: low -> lower.

Author Response

The spelling correction on line 184 has been made as suggested.